# Electrochemical Characterization of Central Action Tricyclic Drugs by Voltammetric Techniques and Density Functional Theory Calculations

**DOI:** 10.3390/ph12030116

**Published:** 2019-08-01

**Authors:** Edson Silvio Batista Rodrigues, Isaac Yves Lopes de Macêdo, Larissa Lesley da Silva Lima, Douglas Vieira Thomaz, Carlos Eduardo Peixoto da Cunha, Mayk Teles de Oliveira, Nara Ballaminut, Morgana Fernandes Alecrim, Murilo Ferreira de Carvalho, Bruna Guimarães Isecke, Karla Carneiro de Siqueira Leite, Fabio Bahls Machado, Freddy Fernandes Guimarães, Ricardo Menegatti, Vernon Somerset, Eric de Souza Gil

**Affiliations:** 1Faculdade de Farmácia, Universidade Federal de Goiás, Goiânia GO 74690-970, Brazil; 2Bloco IQ-1, Campus Samambaia, Avenida Esperança s/n, Instituto de Química, Universidade Federal de Goiás, Goiânia CEP 74690-900, Brazil; 3Department of Chemistry, Faculty of Applied Sciences, Cape Peninsula University of Technology, Bellville 7535, South Africa

**Keywords:** electrochemistry, oxidation mechanism, computational chemistry, electronic structure, cyclobenzaprine

## Abstract

This work details the study of the redox behavior of the drugs cyclobenzaprine (CBP), amitriptyline (AMP) and nortriptyline (NOR) through voltammetric methods and computational chemistry. Results obtained in this study show that the amine moiety of each compound is more likely to undergo oxidation at 1a at *E*_p1a_ ≈ 0.69, 0.79, 0.93 V (vs. Ag/AgCl/KClsat) for CBP, AMP and NOR, respectively. Moreover, CBP presented a second peak, 2a at *E*_p2a_ ≈ 0.98 V (vs. Ag/AgCl/KClsat) at pH 7.0. Furthermore, the electronic structure calculation results corroborate the electrochemical assays regarding the HOMO energies of the lowest energy conformers of each molecule. The mechanism for each anodic process is proposed according to electroanalytical and computational chemistry findings, which show evidence that the methods herein employed may be a valuable alternative to study the redox behavior of structurally similar drugs.

## 1. Introduction

Cyclobenzaprine (CBP), amitriptyline (AMP) and nortriptyline (NOR) are central-acting tricyclic drugs whose applicability in therapy makes them widely prescribed medicines. Although they share similar chemical structures (Figure 1), these drugs display distinct biological activities, with CBP being a muscle relaxant used against acute musculoskeletal pain while AMP and NOR are antidepressants. Regarding CBP, the thermodynamics and kinetics of its redox processes are as yet unknown, so their elucidation is important to better understand the physicochemical features of this drug [1,2,3].

Concerning the study of drug redox features, electrochemical methods such as voltammetry are highly regarded, since both kinetics and thermodynamics are readily assayed using minute solvent volumes. These tools provide information about the redox behavior of the compounds, as well as correlated chemical phenomena [4,5]. Moreover, electrochemistry grants information regarding the oxidative degradation of these compounds [6,7,8].

In addition to redox features acquired by electroanalytical findings, computational chemistry tools such as density functional theory (DFT) calculations may assist in the inference of physical-chemical properties. This approach studies the electronic structure of molecules based on a function of electron density. The optimization of the spatial conformation of each molecule has been performed, and the molecular orbitals energy value difference between each isomer have been calculated [4,8,9]. Such information is applicable to the prediction of possible sites that are susceptible to redox reactions and these data may be correlated to electrochemical findings in order to propose possible oxidation pathways [4,8,9,10,11].

In view of the importance of the elucidation of their redox features, this work is aimed at the characterization of CBP, AMP and NOR through electrochemical and computational chemistry methods [8,9,10,11,12]. Moreover, the correlation between both approaches was determined in order to elucidate and propose a possible electrochemical oxidation pathway of the studied drugs.

## 2. Results and Discussions

### 2.1. Electrochemical Assays

In order to electrochemically characterize CBP, AMP and NOR, a pH study was conducted to investigate the best experimental conditions as well as the dependence of the oxidation process to protons. Thereafter, differential pulse voltammetry (DPV) was further used to evaluate the oxidation of each compound at a fixed pH value, i.e., pH 7.0. Results are depicted in Figure 2.

The results indicate that pH 7.0 provides the best signal amplitude, and this condition apparently makes CBP prone to a second oxidation process (Figure 2A). The pH study for AMP indicated that pH 5.0 provides the best analytical response (Figure 2B), and pH 9.0 is the best for NOR (Figure 2C).

Concerning CBP, AMP and NOR oxidation, the first anodic peak, 1a, seemingly involves the equivalence of protons and electrons, hence the *E*_pa_ vs. pH plot rendered a slope of approximately 59 mV.pH^−1^, which is associated with Nernstian processes (Figure 2). Nonetheless, the literature has reported that proton-dependent electrochemical processes showcase similar slope values in pH studies [4,13,14].

The tricyclic compounds showed similar profiles for cyclic voltammetry (CV) at different scan rates and seemed to have an adsorptive oxidation process, evidenced by the linearity found in *I*_pa_ vs. v (r^2^ = 0.99) (Figure 3A) [4]. This result is also corroborated by the literature [15,16,17] since compounds presenting an abundance of π-electrons tend to adsorb on carbon surfaces, such as that of the carbon paste electrode (CPE) used herein. Furthermore, this behavior is quite similar to that of other tricyclic compounds [13,14,16,18,19].

The square wave voltammetry (SWV) was also similar for all tricyclic compounds and showed no reversibility in the oxidation process (Figure 3B). The composition of differential pulse (DP) voltammograms of each compound at pH 7.0, evidencing their oxidative processes, is depicted in Figure 3C.

DPV findings allowed the determination of the potential associated to the oxidation of each compound. CBP, AMP and NOR showcased the first anodic peak, namely 1a at *E*_p1a_ ≈ 0.69, 0.79, 0.93 V (vs. Ag/AgCl/KClsat), respectively. CBP presented a second peak, 2a at *E*_p2a_ ≈ 0.98 V (vs. Ag/AgCl/KClsat) (Figure 3C).

It can be observed that despite the common dibenzene–cycloheptane core among the tricyclic drugs, they undergo oxidation at different potentials and, thus, their structural differences promote changes in their kinetic behavior.

The SWV results evidenced that the redox processes associated with each anodic peak are irreversible, hence *I*_pa_/*I*_pc_ is higher than one unit, and the CV findings suggested that the process is nonetheless adsorption-controlled [14,18,19].

The next step involved the use of the voltammetric results and electronic structure calculations, in order to propose a possible oxidation pathway.

Therefore, these results were associated to molecular modeling in order to propose a possible oxidation pathway.

### 2.2. Molecular Electronic Structure Calculations

The chemical structures of CBP, AMP and NOR in their lowest energy conformers are presented in Figure 4.

HOMO displays the spread of electronic density in specific regions of the molecule, depicting the likely region of the compound’s first oxidation (Figure 5).

Therefore, the order of *E*_pa_ in anodic processes also follows this trend, with *E*_p1a_ correlating with HOMO. This process was also showcased in a previous outreach by our group [4], as well as in other reports in the literature [20].

In this sense, the amine moiety of each compound is the region in which the HOMO orbital is distributed (Figure 5). This suggests that this group is more likely to be the electro-active site involved in the first anodic process, *E*_p1a_ ≈ 0.85 V (vs. Ag/AgCl/KClsat). It is known that amine moieties usually have high electro-oxidation potentials (i.e., greater than 0.65 V), what corroborates the results depicted herein. The higher potential of peak 1a for NOR may be attributed to its electronic configuration [17,20,21,22,23,24,25,26,27]; nonetheless, the HOMO energy levels also suggested that this region requires a higher overpotential to undergo its redox reaction, whereas CBP and AMP have higher HOMO levels and are thus more easily oxidized (Figure 6).

Orbital energies correlate with the oxidation of compounds according to an electrode’s Fermi level, where higher HOMO values yield anodic processes at lower potentials and vice versa [4]. Furthermore, Mulliken partial charges display electron charge density in different regions of a molecule, which may elucidate regions likely to undergo electro-oxidation in a similar manner to that of HOMO orbitals (Table 1).

The unsaturated C15 (Figure 4) in CBP showed the most negative partial charge value (Table 1). Therefore, CBP C15 is prone to electro-oxidation, being a possible electroactive site responsible for CBP second electro-oxidation.

In order to evaluate the number of electrons in the anodic process of AMP, CBP and NOR, the Laviron equation [1x, 2x] was calculated as follows:$$E_{\mathrm{pa}}=\;E^{\prime }+\frac{2.3\mathrm{RT}}{\left(1-\alpha \right)\mathrm{nF}}\;\mathrm{logv}$$
where *E*_pa_ is the oxidation potential, E’ is the equilibrium potential, R is the universal gas constant, T is the temperature, α is the transference coefficient, n is the number of transferred electrons, F is the Faraday’s constant and v is the scan rate [28,29,30].

Considering α = 0.5 due to the irreversibility of the process [3x], the values of n calculated for AMP, CBP and NOR were 2.3097, 2.5616 and 2.0788, respectively. Thus, the number of electrons involved in the oxidation of AMP, CBP and NOR could be inferred as 2 and, therefore, there was no significant difference in the oxidation kinetics of these drugs.

These findings were used to propose the electro-oxidation pathway depicted in Figure 7.

The proposed electro-oxidation mechanisms contemplate the equivalence of protons and electrons in the first oxidation, i.e., peak 1a. The participation of two electrons (n = 2) in the first oxidation processes, inferred from the Laviron equation, was taken into account in the proposal of the oxidation mechanism (Figure 7). Moreover, the mechanism depicted herein is similar to other reports in the literature concerning the oxidation of aminated tricyclic compounds [25].

Regarding the second oxidation process undergone by CBP (i.e., peak 2a), literature reports have shown that the vinylic bridge at the start of the aliphatic chain C15 (Figure 4) is the most probable site for electron loss. Some reports on forced degradation studies have shown products similar to those of our proposed mechanism [17,20,25,27].

## 3. Materials and Methods

### 3.1. Reagents, Samples and Solutions

All electrolyte salts, solvents and reagents were of analytical grade, and were used without further purification. Electrolyte solutions were prepared with double-distilled Milli-Q water (conductivity ≤0.1 µS cm^−1^; Millipore S. A., Molsheim, France). CBP, AMP and NOR standards (United States Pharmacopea) were used to prepare 1 mmol L^−1^ stock solution immediately prior to the experiments.

### 3.2. Electrochemical Assays

Voltammetric measurements were performed using a potentiostat/galvanostat PGSTAT^®^ model 204 with a FRA32M module (Metrohm Autolab, City, Country) integrated with NOVA 2.1^®^ software. All measurements were performed in a 1 mL one-compartment electrochemical cell coupled to a three-electrode system consisting of a carbon paste electrode (CPE) working electrode prepared with 100 mg of graphite powder and 30 µL of mineral oil, a Pt wire counter electrode and a Ag/AgCl/KCl_sat_ reference electrode (both purchased from Lab solutions, São Paulo, Brazil).

The experimental conditions for differential pulse voltammetry (DPV) were: pulse amplitude of 50 mV, pulse width of 0.5 s and scan rate of 10 mV s^−1^. All voltammetric assays were performed in 0.1 mol L^−1^ acetate buffer solution (ACS), pH 3.0, 5.0 and phosphate buffer solution (PBS), pH 7.0, 9.0 and 11.0. Voltammetric assays were performed in buffers purged with nitrogen gas in order to avoid oxygen interference. The electrochemical area of the carbon paste electrode was 7.77 mm^2^, as estimated in a previous work of our group [31]. The CPE was cleaned and the surface was renewed after each assay.

DP voltammograms were background-subtracted and baseline-corrected to provide better visualization. All experiments were conducted in triplicate and data were analyzed using Origin Pro 9^®^ software (Northampton, MA, USA).

### 3.3. Molecular Electronic Structure Calculations

The computational chemistry protocol used herein was adapted from previous work conducted by our group [4] and consisted of: semi-empiric (PM6, i.e. Parameterization Method 6) and density functional theory (DFT) calculations, which were used for the electronic structure investigation and optimization of CBP, AMP and NOR. A 6-311G++ basis set was employed in DFT. The B3LYP exchange correlation function was used in DFT calculations.

The full geometry optimizations of the molecules were computed considering the two levels of theory applied in this study. The first step in the determination of the most stable molecular geometries was performed by a conformational structural geometry search using the PM6 method, which is much faster than DFT calculation. The lowest energy conformers of CBP, AMP and NOR from the PM6 results were re-optimized by DFT calculations.

For illustration purposes, the orbital distributions of HOMO for AMP and NOR, and those of HOMO/HOMO-1 for CBP were rendered. The distribution of Mulliken charges in each chemical structure was also investigated. The values of LUMO, HOMO, and lower energy occupied orbitals were calculated for all structures. All the calculations were performed using the Gaussian 09 computational chemistry software package [10]. All molecule figures were rendered in Jmol 14 [12].

## 4. Conclusions

The voltammetric and electronic structure calculation findings corroborated the elucidation of CBP, AMP and NOR possible oxidation sites, redox reaction pathway and electrochemical profiling. The results obtained in this study show that the amine moiety of each compound is more likely to undergo oxidation at 1a at *E*_p1a_ ≈ 0.69, 0.79, 0.93 V (vs. Ag/AgCl/KClsat) for CBP, AMP and NOR, respectively. Moreover, CBP presented a second peak, 2a at *E*_p2a_ ≈ 0.98 V, relative to the oxidation of C15 due to the high negative charge observed in Mulliken partial distribution.

Structural changes in compounds with the same core, such as the dibenzene–cycloheptane core, may yield compounds with different redox dynamics. Thus, electrochemical and computational chemistry studies are versatile approaches that can be used in the investigation of redox behavior.

## Figures and Tables

**Figure 1 pharmaceuticals-12-00116-f001:**
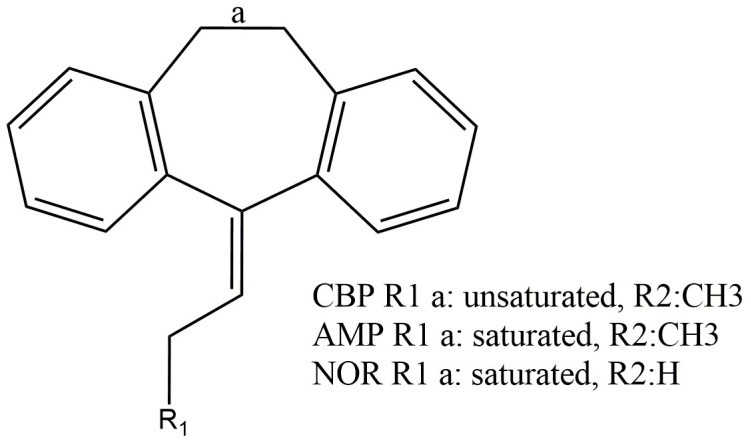
Chemical structure of cyclobenzaprine (CBP), amitriptyline (AMP) and nortriptyline (NOR).

**Figure 2 pharmaceuticals-12-00116-f002:**
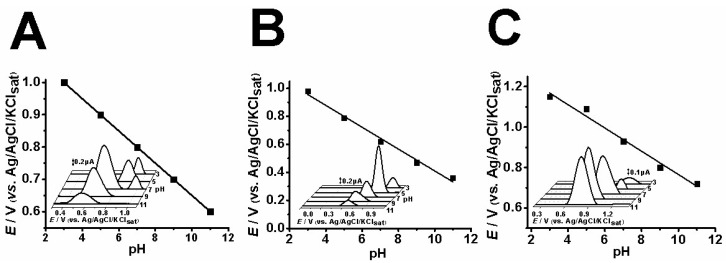
Differential pulse voltammetry results of 0.1 mM CBP (**A**), AMP (**B**) and NOR (**C**) at different pH levels (pH 3, 5, 7, 9, 11) in acetate buffer solution (ABS) and phosphate buffer solution (PBS).

**Figure 3 pharmaceuticals-12-00116-f003:**
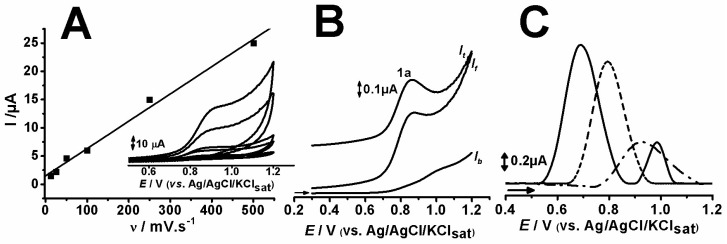
Cyclic voltammetry results at different scan rates (12.5, 25, 50, 100, 250, 500 mV.s^−1^) of 0.1 mM AMP in pH5 ACS (**A**); square wave voltammetry of 0.1 mM AMP in pH5 ACS (**B**); and differential pulse voltammetry of 0.1 mM CBP (—), AMP (- - -) and NOR (- – -) in pH 7 PBS (**C**).

**Figure 4 pharmaceuticals-12-00116-f004:**
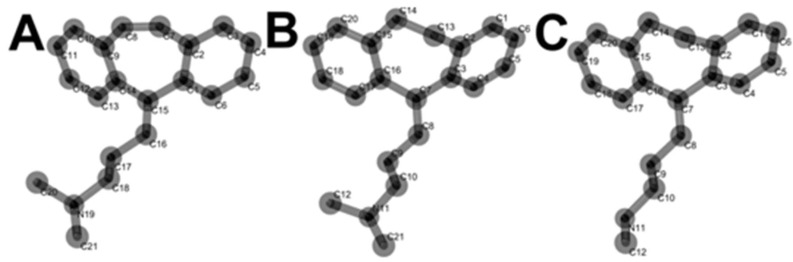
Lowest energy conformers of CBP (**A**), AMP (**B**) and NOR (**C**).

**Figure 5 pharmaceuticals-12-00116-f005:**
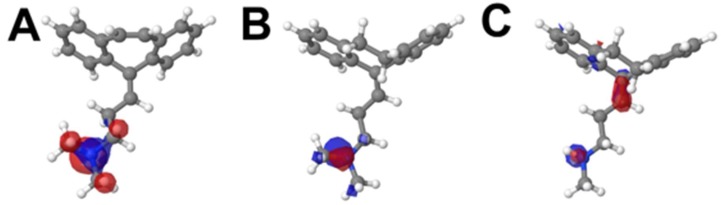
HOMO distribution of CBP (**A**), AMP (**B**) and NOR (**C**), in the molecules’ respective lowest energy conformers.

**Figure 6 pharmaceuticals-12-00116-f006:**
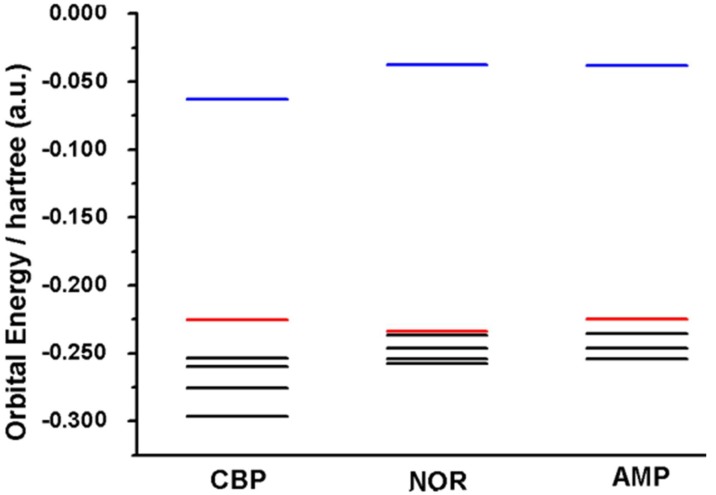
Orbital energy plot of CBP, NOR and AMP, depicting the LUMO (blue line), HOMO (red line) and lower occupied orbitals (black lines).

**Figure 7 pharmaceuticals-12-00116-f007:**
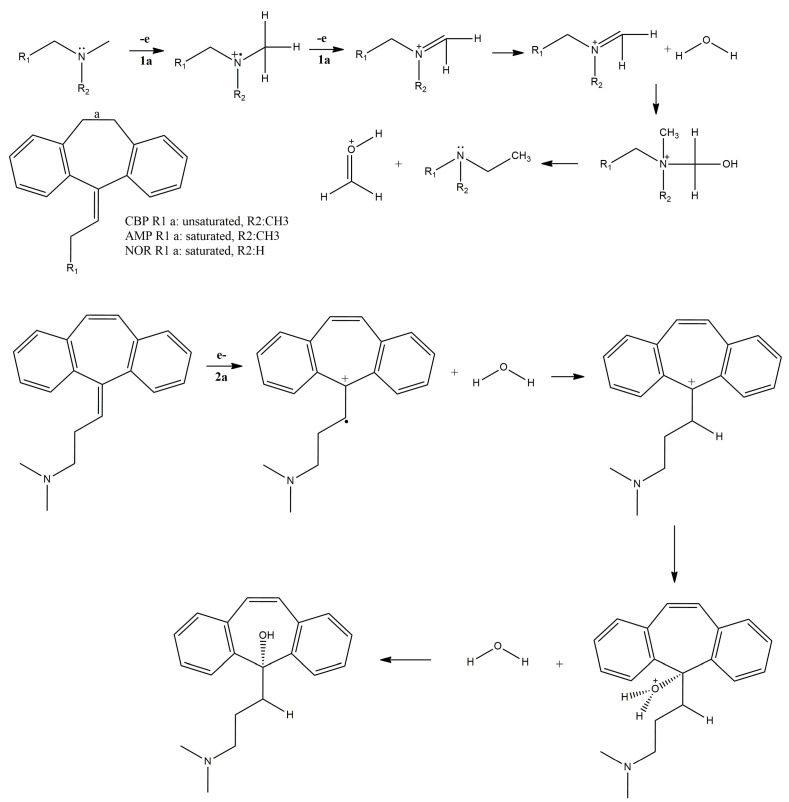
Proposed CBP, AMP and NOR oxidation mechanism.

**Table 1 pharmaceuticals-12-00116-t001:** Mulliken partial charges of CBP, AMP and NOR, represented in percentages of negative and positive charge of each individual atom with hydrogens summed.

CBP	AMP	NOR
Atom Number	Mulliken Charge	Atom Number	Mulliken Charge	Atom Number	Mulliken Charge
C1	−0.042401	C1	−0.112381	C1	−0.119449
C2	0.039541	C2	0.442957	C2	0.447457
C3	−0.027008	C3	0.409320	C3	0.398639
C4	0.010317	C4	−0.221850	C4	−0.235249
C5	0.010578	C5	−0.158772	C5	−0.143385
C6	0.016292	C6	−0.316322	C6	−0.302634
C7	−0.027150	C7	0.996592	C7	0.994888
C8	−0.016568	C8	−0.127010	C8	−0.050367
C9	0.050069	C9	0.080692	C9	−0.001748
C10	−0.032538	C10	−0.584003	C10	−0.490114
C11	0.015741	N11	0.173777	N11	0.121114
C12	0.010368	C12	−0.020840	C12	0.005466
C13	0.021583	C13	0.057774	C13	0.036809
C14	−0.054836	C14	−0.253322	C14	−0.238812
C15	−0.066515	C15	−0.088678	C15	−0.129595
C16	0.112695	C16	0.311043	C16	0.321218
C17	−0.019144	C17	−0.196833	C17	−0.204345
C18	0.142342	C18	−0.174628	C18	−0.128244
N19	−0.416923	C19	−0.415570	C19	−0.424846
C20	0.139806	C20	0.184025	C20	0.143199
C21	0.133751	C21	0.014030

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
