# Peer review of "Electrochemical Characterization of Central Action Tricyclic Drugs by Voltammetric Techniques and Density Functional Theory Calculations"

_pharmaceuticals, 2019, doi:10.3390/ph12030116_

Round 1
Reviewer 1 Report
Review attached.

Author Response
1. From the first sentence of the abstract, it is not clear whether CBP, AMP, and NOR are the names of molecules or solids or biomolecules.
Answer: The names of the molecules were inserted into the text.
2. In Figure 1, what does CIC stand for? Also, insaturated will be unsaturated in the figure.
Answer: CIC was renamed to CBP, which meant cyclobenzaprine and the correction on “unsaturated” was made.
3. On Line 63 of page 2, the statement starting with “and the molecular orbitals energy values between isomers…”, is not clear.
Answer: The text was rewritten to be clearer.
4. On Line 78 of page 3, what is the full form of DPV?
Answer: The text was rewritten to be clearer. DPV stands for differential pulse voltammetry.
5. In Figure 2 caption, what is the full form of ACS and PBS?
Answer: The text was rewritten to be clearer. ACS was renamed to ABS, which stands for acetate buffer solution and PBS stands for phosphate buffer solution.
6. Lines 92-97 on page 3, provide full forms of CV, CPE, and SWV. Give full forms for all the acronyms when they appear the first time in the text.
Answer: The text was rewritten to be clearer.CV stands for cyclic voltammetry, CPE stands for carbon paste electrod and SWV stands for square wave voltammetry.
7. Line 111 on page 4 has an incomplete sentence.
Answer: The setence was completed.
8. Lines 112-118 on page 4: The paragraphs need to be rearranged. For example, the paragraph in Line 112 probably should be moved below the paragraphs following it.
Answer: The text was rewritten to be clearer.
9. In Figure 6, the y-axis needs to be labeled and the units need to be provided.
Answer: The units were added.
10. In Figure 6, HOMO-1 of CBP is more buried than the HOMO-1 of NOR and AMP. Hence, it’s not clear why removal of a second electron from HOMO-1 of CBP occurs (leading to a second peak), but not from HOMO-1 of NOR and AMP. Since the HOMO-1 of CBP is lower in energy than the HOMO-1 of NOR and AMP, one would expect CBP to be less likely than NOR and AMP to give a second peak.
Answer: We apologize for the oversight and we thank the referees for this correction. There are some cases were the HOMO-1 does not reflect the second electro-oxidation, when the molecule undergo unclear changes. The second electro-oxidation proposition is now solely based on the distribution of the Mulliken charges.
11. In Figure 7, insaturated will be unsaturated. Also, N will be a subscript in SN1.
Answer: The change was made.
12. Line 175 of page 8: Some reports in forced degradation studies à Some reports on forced degradation studies
Answers: The corrections were made.
13. Line 221 of page 9: dibenzene-cicloheptane à dibenzene-cycloheptane
Answers: The corrections were made.
Reviewer 2 Report
Manuscript Number: pharmaceuticals-529329
Title: “Electrochemical characterization of central action 3 tricyclic drugs by voltammetric techniques and 4 density functional theory calculations”
Pharmaceuticals
General comments:
This manuscript presents the study of the redox behavior of Cyclobenzaprine (CBP), amitriptyline (AMP) and nortriptyline (NOR) through voltammetric methods and computational chemistry. They then propose a mechanism of oxidation based on their results. Although this is an interesting topic, some aspects should be re-considered by the authors in order to make it acceptable for publication.
Detailed comments format:
1) Figure 1 should be corrected keeping the same abbreviations CBP instead of CIC.
2) The VC and DPV experiments were performed in the presence or absence of oxygen, this must be clarified.
3) The authors must clarify, how was the electrochemical area of their electrodes calculated? Was the electrode subjected to a cleaning treatment between measurements for the VC and DPV experiments?
4) The authors should do a better studyaccording to Laviron, for an adsorption controlled irreversible process, should be calculate the is the transfer coefficient, the reaction heterogeneous electron transfer rate constant and the number of transferred electron. And compare these values with already reported for other molecules
Electroanalysis 2010, 22, No. 19, 2269 – 2276
Electrochimica Acta 56 (2011) 8711– 8717
5) The authors should do a Tafel study to corroborate their proposed mechanism
The work shows results, but there are some doubts about the explanation of the electrochemical responses, suggest doing a detailed study of these electrode for this contribution to the state of the art.
Finally, the language should be checked and revised.
Author Response
Review 2
Figure 1 should be corrected keeping the same abbreviations CBP instead of CIC.
Answers: The corrections were made.
The VC and DPV experiments were performed in the presence or absence of oxygen, this must be clarified.
Answers: The assays were performed without oxygen, this information was added to the methodology section.
The authors must clarify, how was the electrochemical area of their electrodes calculated? Was the electrode subjected to a cleaning treatment between measurements for the VC and DPV experiments?
Answers: This information was added to the text.
4) The authors should do a better study according to Laviron, for an adsorption controlled irreversible process, should be calculate the is the transfer coefficient, the reaction heterogeneous electron transfer rate constant and the number of transferred electron. And compare these values with already reported for other molecules
Electroanalysis 2010, 22, No. 19, 2269 – 2276
Electrochimica Acta 56 (2011) 8711– 8717
Answers: The transference constant was calculated using the Laviron plot. This information was added to the text.
5) The authors should do a Tafel study to corroborate their proposed mechanism
The work shows results, but there are some doubts about the explanation of the electrochemical responses, suggest doing a detailed study of these electrode for this contribution to the state of the art.
Finally, the language should be checked and revised.
Round 2
Reviewer 1 Report
the review is attached as a PDF file.

Author Response
Review 1
(a) “The optimization of a molecule spatial conformation, and the molecular orbitals energy values between isomers can also be obtained with this calculations…”, is still not very clear. It seems the authors want to say: The optimization of the spatial conformation of each molecule was performed, and the molecular orbitals energy value difference between each isomers were calculated.
Answer: The text was rewritten to be clearer.
(b) In Figure 6, the unit was added by the authors for the y-axis. I suggest reporting the values on the y-axis up to 3 decimal places.
Answer: A decimal unit was added to the y-axis
Reviewer 2 Report
THE CORRECTIONS WERE CARRIED OUT.
NEW REFERENCES MUST BE REVISED WITH THEIR CORRESPONDING NUMBERS.Author Response
We thank the referees for their attetion and valuable insights to our work. The correction and suggestions are appreciated.
Review 1
(a) “The optimization of a molecule spatial conformation, and the molecular orbitals energy values between isomers can also be obtained with this calculations…”, is still not very clear. It seems the authors want to say: The optimization of the spatial conformation of each molecule was performed, and the molecular orbitals energy value difference between each isomers were calculated.
Answer: The text was rewritten to be clearer.
(b) In Figure 6, the unit was added by the authors for the y-axis. I suggest reporting the values on the y-axis up to 3 decimal places.
Answer: A decimal unit was added to the y-axis
Review 2
NEW REFERENCES MUST BE REVISED WITH THEIR CORRESPONDING NUMBERS.
Answer: References were reviewed.